# OpenReview forum: "MIRAGE-Bench: LLM Agent is Hallucinating and Where to Find Them"
_ICLR.cc/2026/Conference — Submitted to ICLR 2026_

### Official Review · Reviewer_vWvc · 2025-10-28

**Soundness:** 2
**Presentation:** 2
**Contribution:** 2
**Rating:** 4
**Confidence:** 3

**Summary:**

This paper proposes MIRAGE-Bench, a unified benchmark for studying hallucinations in agentic language models. It uses a three-part taxonomy, a set of contextual “snapshot” test cases, and an LLM-as-a-Judge evaluation scheme. The goal is to provide a principled testbed for analyzing hallucinations in interactive agents. Extensive experiments across multiple environments reveal widespread unfaithfulness among current agents. The results also show that proprietary and open-source models perform similarly, indicating that scaling alone cannot guarantee faithfulness.

**Strengths:**

1. The paper introduces the first single benchmark, MIRAGE-BENCH, and a clear, three-part categorization to systematically study and evaluate when and why LLM agents hallucinate.

2. To solve the problem of unpredictable agent behavior in dynamic environments, the authors use a new contextual snapshot strategy to reliably repeat and test agent decisions at specific failure points.

3. The research goes beyond simple scoring to analyze why hallucinations happen, revealing that agents often fail because their training data is too focused on "successful workflows," causing them to ignore critical error feedback.

**Weaknesses:**

1. The LLM-as-a-Judge setup limits the reliability of evaluation. Validation is based on only 160 human-labeled samples with moderate agreement, which is insufficient to ensure trustworthiness. Relying on one LLM to judge another introduces unverified bias and instability, especially under prompt variations.

2. The Contextual Snapshot Strategy sacrifices dynamic fidelity for reproducibility. By freezing the agent’s state before potential hallucination points, the benchmark reduces complex multi-turn reasoning to isolated steps. It therefore fails to capture long-horizon planning, feedback integration, and recovery abilities crucial for real-world agents.

3. The paper diagnoses a key “successful workflow” bias but lacks an effective mitigation. While the analysis convincingly links hallucination to overfitting on optimal trajectories, it offers no concrete or tested method to reduce this bias. Merely calling for future work on “risk settings” leaves the contribution incomplete.

4. The benchmark’s generalizability is limited by dependence on structured environments. Most data come from existing benchmarks with structured HTML trees or terminal outputs, making some risk types (e.g., Pop-up Distractions) ineffective. It underrepresents hallucinations in unstructured text, documents, or visual contexts found in generalist agents.

5. The conceptual boundary between hallucination and general error remains unclear. Many reported hallucinations could be reframed as planning or attention failures. This ambiguity weakens the core claim that scaling offers little gain in faithfulness, as results may reflect labeling uncertainty rather than genuine performance limits.

**Questions:**

Please refer to the above-mentioned weaknesses.

---

> ### Author Response · Authors · 2025-11-21
> **Response (Part 1)**
>
> We thank the reviewer for the extensive and thought-provoking feedback! We appreciate the opportunity to clarify the rigor of our evaluation protocols, and we value the stimulating discussions raised regarding the conceptual boundaries and scope of agentic hallucination.
>
> ### W1: Concerns on LLM-as-a-Judge Reliability
>
> We respectfully clarify that our evaluation is grounded in a statistically rigorous human baseline and has passed multi-dimensional robustness checks (human alignment, cross-model validation, and prompt sensitivity), as summarized in Lines 401–408 and detailed in Appendix C.2.
>
> 1. **Rigor of Human Ground Truth (Sample Size & Agreement):** Contrary to the concern about sufficiency, our human evaluation is statistically robust:
>     - **Statistical Validity:** The 160 samples represent **~15.2%** of the dataset. We utilized **stratified sampling** (covering every risk setting and score distribution) rather than simple random sampling. Statistically, this yields a conservative margin of error of **$\pm 7\%$** (95% confidence interval), well within standard NLP norms.
>     - **Expertise & Agreement**: Annotators were Computer Science students trained on the paper’s definitions. To further validate reliability, two co-authors independently re-annotated the validation set, with each expert covering 80 samples to collectively cross-verify the entire 160-sample subset against the student labels. We report a **Quadratic Weighted Kappa of 0.818**, indicating substantial agreement and confirming high consistency in the ground truth.
>
> 2. **Mitigating Bias: Human-Aligned Cross-Validation:** We did not rely solely on one model. We benchmarked **o4-mini, Claude-3.5-Sonnet, and Gemini-1.5-Flash** against the human ground truth. As shown below, all models achieved high accuracy (>0.75), proving that the judge's performance is not a result of specific model bias but aligns with human reasoning.
>
> | LLM Judge | Accuracy | ZeroAcc |
> | :--- | :--- | :--- |
> | **o4-mini** | 0.756 | 0.789 |
> | **Claude-3.5-Sonnet** | 0.769 | 0.895 |
> | **Gemini-1.5-Flash** | 0.769 | 0.806 |
>
> 3. **Addressing Instability: Sensitivity to Prompt & Temperature:** We explicitly tested the "instability under prompt variations" concern raised by the reviewer through two stress tests.
>     - **Prompt Format Sensitivity:** We randomized the structural order of prompt components (instructions, context, criteria). Despite these significant perturbations, agreement remained **>0.77**, and notably, **ZeroAccuracy** (hallucination detection) consistently exceeded **0.85**.
>     - **Self-Consistency:** We assessed stability by running repeated trials at temperature=1. Agreement scores consistently exceeded **0.8** across trials.
>
> | Validation Experiments                 | Accuracy | ZeroAcc |
> |----------------------------------------|----------|---------|
> | **Robustness to Prompt Format Variations**         |          |         |
> | o4-mini                                | 0.756    | 0.789   |
> | Claude-3.5-Sonnet                      | 0.769    | 0.895   |
> | Gemini-2.5-flash                       | 0.769    | 0.806   |
> |                                        |          |         |
> | **Robustness to Sampling Temperature** |          |         |
> | o4-mini-run1                           | 0.849    | 0.845   |
> | o4-mini-run2                           | 0.819    | 0.831   |

---

> ### Author Response · Authors · 2025-11-21
> **Response (Part 2)**
>
> ### W2: Contextual Fidelity vs. Trajectory Continuation
>
>
> We clarify that snapshots preserve full multi-turn reasoning fidelity in the input, while the lack of subsequent interaction is an acknowledged limitation mitigated by our granular scoring design.
>
> 1.  **Preserved Multi-Turn Fidelity:** Contextual snapshots do **not** sacrifice dynamic fidelity regarding the input. As detailed in Section 4.2, each snapshot is extracted from a realistic multi-turn trajectory where environment and task states have been **dynamically evolved** by prior steps. The LLM receives the **full interaction history** and current observation, ensuring it performs reasoning over a valid, long-horizon context.
> 2.  **Acknowledged Limitation & Scoring Mitigation:** We openly acknowledge that by freezing the interaction state to ensure reproducibility, we sacrifice the **subsequent trajectory execution**. Consequently, we cannot dynamically observe whether an ambiguous action (e.g., `WAIT`, `SCROLL`) successfully retrieves useful feedback or leads to recovery in the following steps. To address this, we utilize the **intermediate Score 1**, ensuring that such conservative, utility-bearing actions are not penalized as hallucinations (Score 0), effectively isolating faithful uncertainty from unfaithful fabrication.
> 3. **Evaluation of Core Capabilities via Risk Settings:**
> Our benchmark evaluates whether the agent possesses capabilities like planning, recovery, and feedback integration sufficiently to remain **faithful** under realistic risks, rather than measuring them via generic task completion. Far from being ignored, these abilities are inherently stress-tested by our specific risk settings:
>     * **Recovery** is explicitly evaluated in *Flawed Interaction History*, where the agent must identify and correct previous errors encoded in the context.
>     * **Feedback Integration** is the core challenge of *Unexpected Environmental Transitions*, testing if the agent can adapt to observations that contradict its internal prediction.
>     * **Long-Horizon Planning** is implicit in every sample, as snapshots are taken from deep within multi-step trajectories requiring context-dependent decision-making.
>
> ### W3: Absence of Mitigation for "Successful Workflow" Bias
>
>
> Our primary contribution is establishing the first rigorous evaluation framework for agent hallucination, which serves as the necessary prerequisite for developing effective solutions.
>
> 1.  **Foundational Diagnosis:** The identification of the "successful workflow" bias is just one of several key insights enabled by this protocol. By providing the first rigorous protocol to **elicit and measure** diverse hallucination patterns, we transform them from hypotheses into quantifiable metrics. Without this diagnostic benchmark, any proposed mitigation—for this bias or other risks—would lack a grounded standard for evaluation.
> 2.  **Future Roadmap:** We agree that mitigation is the imperative next step. Building on this framework, we are already investigating strategies to counteract these risks. This work provides the essential empirical groundwork upon which such solutions can be systematically developed.
>
>
> ### W4: Generalizability and Reliance on Structured Environments
> We prioritize the realistic interfaces of deployed digital agents, where structured inputs are standard, while ensuring interpretability over visual opacity.
>
> 1.  **Realistic Deployment Standard:** Current mature digital agents (coding, web navigation, enterprise automation) fundamentally rely on structured interfaces (HTML, DOM trees, Terminals) to operate reliably. Our benchmark reflects this **real-world deployment reality**, covering the most diverse and representative domains available today.
> 2.  **Interpretability vs. Visuals:** We prioritize text-centric settings to enable precise **root-cause analysis** (e.g., attributing "cognitive inertia"). Visual inputs, while important for future work, currently introduce opacity that hinders the granular diagnosis of hallucination mechanics.

---

> ### Author Response · Authors · 2025-11-21
> **Response (Part 3)**
>
> ### W5: Conceptual Boundary between Hallucination and General Error
>
> We distinguish hallucination as a specific subset of "wrong actions" characterized by unfaithfulness. We illustrate this distinction and its implication for training using a typical case:
>
> 1.  **Distinction via Case Study:** Consider a non-hallucinatory wrong action where an agent navigates to an incorrect page due to a lack of domain knowledge. Although the action is "wrong," the agent remains **faithful to the observation** by realizing the mismatch and stopping or attempting recovery. In contrast, a hallucination occurs when the agent performs the *same* wrong action but ignores the contradictory observation, **overconfidently acting** as if it were correct (e.g., clicking a non-existent button). This specific "detachment from reality"—not the mistake itself—defines the boundary we rigorously label.
>
> 2.  **Implications for Scaling & Training:** This distinction clarifies why scaling may offers limited gains. While post-training injects knowledge to reduce wrong actions (e.g., learning the correct path), it does not guarantee the agent will remain **faithful** when facing novel environments. Paradoxically, as analyzed in our "Cognitive Inertia" section, overfitting to "successful workflows" can **worsen** faithfulness: 'stronger' models may prioritize their learned internal plans over subtle but critical contradictions in real-time observations, leading to confident hallucinations rather than honest failure.

---

### Official Review · Reviewer_GBRL · 2025-10-30

**Soundness:** 3
**Presentation:** 3
**Contribution:** 3
**Rating:** 6
**Confidence:** 4

**Summary:**

MIRAGE-Bench introduces the first unified benchmark for hallucinations in interactive LLM agents. It proposes a three-part pipeline: (1) a 3-way taxonomy (unfaithful to instructions / history / observations); (2) 6 risk settings + contextual snapshot freezing to elicit reproducible hallucinations; (3) risk-specific LLM-as-a-Judge for scalable action-level verification.

**Strengths:**

+ Fills a critical gap: First systematic benchmark for interactive agent hallucinations—beyond single-turn QA (TruthfulQA, HaloGEN) and success-only agent evals (WebArena, AgentBench). Table 1 clearly shows missing dimensions.

+ Strong taxonomy: Grounded in ReAct loop; each category maps to real-world risks (e.g., credential leak via fake navigation, Fig 2).

+ Snapshot innovation: Freezing full context (instruction + history + observation) at hallucination-prone steps eliminates stochasticity while preserving multi-turn complexity. Enables environment-free, reproducible testing.

+ Smart positive design: Treats "acknowledge uncertainty / refuse / report infeasibility" as faithful behavior (e.g., Out of Scope Queries)—a safety-minded shift from "always answer" paradigms.

**Weaknesses:**

- Human evaluation critically under-specified: Only 160 samples used for judge validation. No annotator expertise reported (e.g., agent safety researchers?). No inter-annotator agreement (Cohen’s κ, Krippendorff’s α). If more detailed information such as a substantially larger validation set documented domain expertise of raters and published inter-rater reliability scores were provided the credibility of the AI-safety assessment would be significantly strengthened.

- Multi-turn" claim vs. snapshot reality: Snapshots are static slices of multi-turn trajectories. They test single-step faithfulness under long context, not dynamic accumulation of hallucinations over turns. Misalignment with paper’s framing as a “multi-turn hallucination” benchmark.

- Analysis depth missing: No correlation studies: model size vs. hallucination type, snapshot depth vs. error rate, risk complexity vs. failure. No ablation on judge prompt design, snapshot selection criteria, or error cases.

- Dataset transparency weak: 1,050 samples unevenly distributed (8.4%–22.1%). No per-environment breakdown, no raw trajectory release.

**Questions:**

See weakness.

---

> ### Author Response · Authors · 2025-11-21
> **Response (Part 1)**
>
> We sincerely thank the reviewer for the meticulous and insightful feedback! Your constructive comments, particularly regarding human evaluation rigor and analysis depth, have helped us significantly strengthen the empirical validity and transparency of our work.
>
>
> ### **W1: Human Evaluation Rigor & Inter-Annotator Agreement**
>
> We employed statistically significant stratified sampling and have now added inter-annotator agreement metrics to validate human ground truth.
> 1. **Statistical Validity of Sample Size**: The 160 samples represent **~15%** of the full dataset (1,050), yielding a conservative margin of error of $\pm 7\%$ (95% confidence interval, assuming maximum variance with finite population correction). What's more, unlike simple random sampling, our stratified approach enforces balanced coverage of score distributions (Scores 0/1/2) and all risk settings. This serves as a rigorous stress-test on ambiguous boundary cases to ensure unbiased evaluation.
> 2. **Annotator Expertise**: Annotators were Computer Science students who underwent a rigorous training protocol: reviewing the paper’s formal definitions, analyzing illustrative cases per risk setting, and studying the judge prompt criteria to ensure strictly aligned domain understanding.
> 3. **Inter-Annotator Agreement**: As suggested by the reviewer, we further strengthened the credibility of our human evaluation. Two co-authors independently re-annotated the validation set, with each expert covering 80 samples to collectively cross-verify the entire 160-sample subset against the student labels. Given the ordinal nature of our scoring system (Scale 0 $\to$ 1 $\to$ 2), we calculated the **Quadratic Weighted Kappa**. The resulting score of **0.818** indicates substantial agreement, validating the high consistency and reliability of our human ground truth.
> 4. **Multi-Dimensional Computational Validation**: Beyond human verification, we reinforce credibility through comprehensive computational validation (summarized in Lines 401–408, detailed in Appendix C.2):
>     - **Human-Aligned Cross-Validation**: We benchmarked o4-mini, Claude-3.5, and Gemini against the human ground truth. All models achieved high accuracy (>0.75), confirming strong alignment.
>     - **Self-Consistency**: We validated the stability of our main evaluator (o4-mini) across repeated runs (temp=1), yielding agreement scores consistently >0.8.
>     - **Prompt Format Sensitivity**: We stress-tested the judge by randomizing the order of prompt components. Despite structural perturbations, accuracy remained >0.77 (with hallucination-specific accuracy >0.85), proving the stability across different prompt formats.
>
> ### W2: Clarification on "Multi-turn" Framing vs. Snapshot Evaluation
>
> Our benchmark evaluates decisions conditioned on multi-turn history to ensure clean attribution, focusing on the origin of hallucination rather than its dynamic accumulation.
>
> 1. **Multi-turn Context as Input**: The "multi-turn" designation refers to the data source and input complexity. Each snapshot captures a state evolved through realistic interactions, requiring the agent to reason over a long-horizon history and accumulated observations, rather than simple single-turn instructions.
> 2. **Precision vs. Compounding Errors**: We intentionally isolate the first risky decision point. Evaluating dynamic accumulation introduces compounding errors, making it difficult to distinguish between a new hallucination and the cascading effects of a previous one. Our snapshot approach guarantees that any failure is a direct hallucination at that specific critical step.
> 3. **Scope Clarification and Future Research**: We strongly agree that studying error accumulation and self-recovery is a valuable complementary direction. We will revise the text to explicitly frame our work as diagnosing the root cause in long contexts, leaving dynamic error evolution for future research.

---

> ### Author Response · Authors · 2025-11-21
> **Response (Part 2)**
>
> ### W3: Analysis on Correlations and Methodological Ablations
> Below, we address the specific correlation and ablation inquiries.
>
> 1.  **Correlation Study - Snapshot Depth vs. Error Rate:**
>     We analyzed the impact of interaction length (Depth 4 vs. 7) in the *Flawed Interaction History-repetitive* setting across 17 models ($N=3049$). The analysis reveals a **negligible effect**: the mean difference in judge scores is only $\Delta = -0.092$, with a **Cliff’s delta of 0.053** and point-biserial correlation of $r = -0.050$. This indicates no systemic evidence that "deeper steps" monotonically increase hallucination rates; rather, failures are driven by complex latent factors beyond simple interaction length.
> 2.  **Methodological Ablations & Error Analysis:**
>     * **Judge Prompt Design:** The final design was reached through iterative ablation: we found that adding explicit explanations of the risk setting significantly improved the judge's understanding of hallucinated actions, whereas including few-shot examples paradoxically degraded generalization to diverse cases. We also ablate the judge prompt design via format perturbation, as detailed in response on W1.
>     * **Selection & Error Taxonomy:** Due to space constraints, the specific snapshot selection pipeline for each risk setting, along with a detailed taxonomy of model decision patterns and error cases, are comprehensively reported and analyzed in Appendix B. We thank the reviewer for highlighting the importance of these details and welcome interested researchers to leverage these analyses for further investigation.
> ### W4: Dataset Imbalance and Trajectory Availability
>
> We appreciate the reviewer for highlighting these statistics and acknowledge the uneven distribution. We clarify that this reflects the inherent trade-off between construction complexity and environment scope, while transparency is ensured through comprehensive log releases.
>
> 1. **Rationale for Distribution**: Sample volume is dictated by construction complexity. Risk settings requiring manual annotation to pinpoint precise risk steps naturally yield fewer samples than those amenable to heuristic automation. We are actively working on balancing this in future versions.
> 2. **Per-Environment Breakdown**: Data volume correlates with environment flexibility. WebArena (31.9%) supports diverse, open-ended scenarios allowing broad risk instantiation, whereas TauBench (9.8%) is strictly constrained to vertical domains (retail/airline), limiting the design space.
>
> | Environment     | Percentage |
> |-----------------|------------|
> | WebArena        | 31.9%      |
> | SWE-Bench       | 20.9%      |
> | WorkArena       | 19.4%      |
> | TheAgentCompany | 18.0%      |
> | TauBench        | 9.8%       |
>
> 3. **Transparency & Resources**: Raw trajectories undergo heavy filtering to isolate specific risk conditions, reducing their standalone value. Instead, we ensure transparency by releasing all test results and detailed LLM judge results at [an anonymized repository](https://anonymous.4open.science/r/mirage-bench-exp-results-2BC8/README.md), enabling researchers to conduct granular further studies.

---

### Official Review · Reviewer_kAje · 2025-11-01

**Soundness:** 3
**Presentation:** 3
**Contribution:** 3
**Rating:** 4
**Confidence:** 3

**Summary:**

This paper introduces MIRAGE-Bench, a unified benchmark for eliciting and evaluating hallucinative actions in LLM-based agents. The authors define a three-part taxonomy of unfaithful behaviors (to task instructions, interaction history, and environment observations) and propose a snapshot elicitation strategy that freezes risky decision points for deterministic and reproducible evaluation. Furthermore, a risk-aware LLM-as-a-Judge protocol labels each action as faithful, incomplete, or hallucinative to derive Utility Scores (US) and Hallucination Rates (HR). Experiments span diverse environments, including web, operating systems, software engineering, and inter-agent tasks, revealing that hallucinations persist even in strong proprietary models.

**Strengths:**

(1) This paper formally defines hallucinative actions and distinguishes three types of unfaithful behaviors (task instructions, interaction history, and environment observations) thereby extending the notion of hallucination from natural language generation to action-level decision-making in interactive agents.

(2) The paper clearly defines key concepts such as hallucinative actions, the snapshot strategy, and the risk-aware LLM-as-a-Judge framework, and presents a logical and easy-to-follow flow from motivation to methodology to results. The writing is concise, terminologically consistent, and technically transparent, making the presentation clear and accessible.

(3) The study presents a well-structured experimental design covering six representative risk scenarios. Its risk-aware LLM-as-a-Judge framework with three-way classification (faithful / incomplete / hallucinative) enables fine-grained evaluation via Utility Score (US) and Hallucination Rate (HR).

**Weaknesses:**

(1) The benchmark focuses on six well-chosen but mainly text- and web-centric settings. Including non-web domains (e.g., embodied or multimodal agents) would improve generality and demonstrate broader applicability.

(2) Although the snapshot strategy ensures reproducibility, the paper does not show whether snapshots preserve full contextual fidelity.
Experiments comparing snapshot vs. full-trajectory evaluation or perturbation tests would better support this assumption.

**Questions:**

Please see Weaknesses

---

> ### Author Response · Authors · 2025-11-21
> **Response**
>
> We thank the reviewer for raising questions regarding domain generality and snapshot fidelity. Below, we clarify the rationale behind our focus on realistic digital agents and explain how fidelity is structurally guaranteed by our design.
>
> ### W1: Generality and Inclusion of Non-Web Domains
>
> We prioritize mature digital agents to ensure grounded risk assessment and precise interpretability.
> 1. **Broad Digital Domain Coverage**: Our benchmark already extends far beyond simple text tasks, covering the most diverse and complex domains available for current agents: open web (WebArena), enterprise management (WorkArena, TheAgentCompany), professional coding (SWE-Bench), and vertical domain services (TauBench).
> 2. **Focus on Realistic, Near-Deployment Settings**: While embodied environments such as AlfWorld, OmniGibson, and Habitat provide useful simulation testbeds, the way language models are engaged in these settings is still highly exploratory: the tasks and environments are heavily simplified, the interfaces to the world are largely simulator-specific, and real-world deployment of such embodied agents remains very limited. In contrast, digital LLM agents that operate via APIs and function calls in text- and web-based environments already constitute mature and widely deployable use cases. We believe systematically characterizing their real-world risks is more grounded and urgent at this stage.
> 3. **Clarity & Tractability vs. Multimodal**: While we acknowledge the potential of multimodal GUI agents, text-based settings are clearer, lightweight, and easier to analyze. For instance, our key insight on "cognitive inertia" was derived via attribution analysis on text subsequences—a process that remains challenging and opaque with heavy visual encoders. Our work lays the foundational methodology for future multimodal extensions.
>
>
> ### W2: Contextual Fidelity of Snapshots
> Contextual fidelity is structurally guaranteed by design, not an assumption.
>
> As illustrated in Section 4.2 and Figure 3, a snapshot is **not** a reconstruction or approximation. It is the **exact, bit-wise replica** of the full context window (task instruction + complete interaction history + current observation) fed to the agent at the specific risk step during a dynamic rollout. Since we construct snapshots by simply "pausing" the full trajectory and saving the state, the model receives identical input in our snapshot evaluation as it would in a full-trajectory run. Therefore, contextual fidelity is preserved 100% by definition, rendering comparison experiments redundant.

---

### Official Review · Reviewer_8RnA · 2025-11-02

**Soundness:** 2
**Presentation:** 2
**Contribution:** 2
**Rating:** 4
**Confidence:** 3

**Summary:**

This paper introduces a unified benchmark, MIRAGE-Bench, designed to evaluate hallucinations in LLM agents. The authors propose a three-part taxonomy of agentic hallucinations, defined by unfaithfulness to task instructions, interaction history, or environment observations. The benchmark leverages a contextual snapshot strategy to isolate and reproduce hallucination-prone decision points across multiple environments and tasks. Evaluation is conducted through LLM-as-a-Judge, which enables scalable and fine-grained assessments. Through quantitative and qualitative analyses across twelve open-source and proprietary models, the study reveals the pervasiveness of hallucinations and argues that they are not mitigated by scale or model size alone.

**Strengths:**

1. This paper proposes a unified taxonomy of agentic hallucinations that categorizes failures based on unfaithfulness to task instructions, interaction history, and environmental observations.
2. The contextual snapshot strategy addresses non-determinism and setup complexity of full environments, which enables stable and reproducible evaluations without requiring full environment rollouts.
3. The benchmark covers a diverse range of interactive environments, spanning web, OS, software-engineering, and task-oriented multi-agent contexts.

**Weaknesses:**

1. Relies solely on Claude-3.5-Sonnet as the judge model, which may introduce bias or limit generalizability. The evaluation would be more robust with cross-validation using multiple judge models (e.g., GPT, Gemini) or ablation on judge sensitivity.
2. More advanced state-of-the-art LLMs such as GPT-5, Gemini 2.5 Pro, and Claude-4-Sonnet/Opus are not evaluated. Would models with larger reasoning abilities alleviate agentic hallucinations? The paper lacks an ablation study on models with varying reasoning capabilities.
3. The authors provide some analyses but stop short of proposing of evaluating some concrete mitigation strategies beyond a vague call for "training on risk contexts.

**Questions:**

Please see the weakness section above

---

> ### Author Response · Authors · 2025-11-21
> **Response**
>
> We thank the reviewer for the constructive feedback, particularly the valuable suggestion to evaluate the latest SOTA models which has significantly strengthened our analysis, and the opportunity to further clarify the robustness of our evaluation framework.
>
> ### W1: Robustness of LLM-as-a-Judge
>
> We respectfully clarify that we have conducted a comprehensive multi-dimensional validation of our judge, including human alignment, self-consistency, and prompt sensitivity tests in our paper(Lines 401–408, Appendix C.2).
>
> 1. **Human-Aligned Cross-Validation**: We did not rely on a single model. We benchmarked o4-mini, Claude-3.5-Sonnet, and Gemini against human annotations (ground truth) on a balanced 160-sample subset. As shown below, all models achieved high accuracy (>0.75), validating the reliability of our judge against human judgment.
>
> | LLM Judge            | Accuracy | ZeroAcc |
> |----------------------|----------|---------|
> | o4-mini              | 0.756    | 0.789   |
> | Claude-3.5-Sonnet    | 0.769    | 0.895   |
> | Gemini-2.5-Flash     | 0.769    | 0.806   |
>
> 2. **Self-Consistency**: We further assessed the stability of our main evaluator (o4-mini, temp=0) by comparing it against repeated runs at temperature=1. The results confirm high self-consistency, with agreement scores consistently exceeding 0.8 across trials.
>
> | Temp=1                 | Accuracy | ZeroAcc |
> |------------------------|----------|---------|
> | o4-mini-run1           | 0.849    | 0.845   |
> | o4-mini-run2           | 0.819    | 0.831   |
>
> 3. **Sensitivity to Prompt Format**: Aligned with prior robustness studies[A, B], we stress-tested the judge by randomizing the structural order of prompt components (e.g., instructions, context, criteria). Despite these significant structural perturbations, the table below shows that agreement with the original format remained >0.77, and notably, ZeroAccuracy (specific to hallucination detection) consistently exceeded 0.85.
>
> | Comparison                    | Accuracy | ZeroAcc |
> |-------------------------------|----------|---------|
> | format_1 vs original    | 0.875    | 0.950   |
> | format_2 vs original    | 0.775    | 0.850   |
>
> 4. **Qualitative Validation**: Beyond metrics, Table 4 provides case studies demonstrating that the judge model offers differentiated scores with accurate, interpretable justifications.
>
> [A] Ghazarian, Sarik, et al. "Assessment and Mitigation of Inconsistencies in LLM-based Evaluations." (2024).
>
> [B] Ngweta, Lilian, et al. "Towards LLMs robustness to changes in prompt format styles." (2025).
>
>
> ### W2: Evaluation on Latest SOTA Models
>
> Stronger reasoning capabilities mitigate but do not eliminate agentic hallucinations; even a reduced rate remains a critical bottleneck for reliable deployment.
>
> 1. **Updated Performance**: We conducted additional evaluations on the latest proprietary models according to the reviewer's suggestion. As shown in the table below (and updated Table 3), Claude-4.5-Sonnet achieves a hallucination rate of 0.154, significantly outperforming the previous best of 0.290.
>
> | Comparison              | Overall Utility Score (US) ↑ |Overall Hallucination Rate (HR)↓ |
> |-------------------------|----------|---------|
> | Claude-4.5-sonnet | 0.712    | 0.154   |
> | Gemini-2.5-pro    | 0.632    | 0.269   |
> | GPT-5             | 0.617    | 0.267   |
>
> 2. **Significance of Residual Risk**: Despite this improvement, a 15.4% hallucination rate remains prohibitively high for practical adoption. Unlike isolated QA tasks, agentic workflows often involve thousands of executions where errors accumulate and amplify over long horizons. This persistence of hallucination, even in the most advanced models, underscores the urgency of our benchmark and the need for dedicated mitigation strategies.
>
>
> ### W3: Absence of Concrete Mitigation Strategies
>
> Our primary contribution is establishing the first rigorous evaluation framework for agent hallucination, which serves as a necessary prerequisite for developing effective mitigations.
>
> 1. **Foundational Scope**: To the best of our knowledge, this is the first work to formalize hallucination specifically for LLM agents. While prior research largely focuses on capability limits, we prioritize faithfulness and robustness of multi-turn interaction. Establishing this grounded definition and reproducible protocol is the necessary foundation before any mitigation can be accurately assessed.
> 3. **Future Roadmap**: We agree that mitigation is the critical next step. Building on the framework introduced here, we are already investigating both training-based and training-free approaches. We view this current work as providing the essential conceptual and empirical groundwork upon which these techniques can be systematically developed and compared.

---

### Author Response · Authors · 2025-11-21
**Summary of response -- we thank all the reviewers for their constructive and valuable feedback.**

We are encouraged that the reviewers recognize the significance of MIRAGE-Bench as the first unified benchmark to systematically evaluate hallucinations in interactive agents (R3, R4). All reviewers unanimously praised our three-part taxonomy of agentic hallucinations (unfaithfulness to instructions, history, and observations) as a strong and grounded framework (R1, R2, R3, R4), with R2 noting that it successfully extends the concept of hallucination from natural language generation to action-level decision-making. The contextual snapshot strategy was widely highlighted as a key innovation that addresses the stochasticity of dynamic environments to enable reproducible and stable evaluation (R1, R2, R3, R4). Furthermore, reviewers acknowledged the value of our comprehensive experimental design, specifically the coverage of diverse risk scenarios and environments (R1, R2), the "safety-minded" metrics that reward acknowledging uncertainty (R3), and our insightful diagnosis that current agents suffer from a "successful workflow" bias (R4).

We have addressed the reviewers’ comments and concerns in individual responses to each reviewer.

(\* As abbreviations, we refer to **Reviewer 8RnA** as R1, **Reviewer kAje** as R2, **Reviewer GBRL** as R3，and **Reviewer vWvc** as R4 respectively.)

---

### Meta-Review · Area_Chair_ttMT · 2025-12-24

**Summary:**

While reviewing this submission, Reviewer kAje found that the authors’ anonymous repository link (https://anonymous.4open.science/r/mirage-bench-250E/README.md) reveals identifiable author information in 01 Nov 2025. Therefore, this work should be desk-rejected considering it violates the double-blind rule required by ICLR 2026.

**Reviewer Concerns:**

Reviewer kAje found that the authors’ anonymous repository link (https://anonymous.4open.science/r/mirage-bench-250E/README.md) reveals identifiable author information in 01 Nov 2025. Then this work violates the double-blind rule required by ICLR 2026 and should be desk-rejected.

**Reviewer Scores:**

It is serious for the authors violate the double-blind rule required by ICLR 2026 considering that the authors’ anonymous repository link (https://anonymous.4open.science/r/mirage-bench-250E/README.md) reveals identifiable author information pointed by Reviewer kAje in 01 Nov 2025. Then all comments and scores from reviewers are invalid considering this reason.

---

### Decision · Program_Chairs · 2026-01-26

Reject